# Saliency-Aware Subtle Augmentation Improves Human Visual Search Performance in VR

**DOI:** 10.3390/brainsci11030283

**Published:** 2021-02-25

**Authors:** Olga Lukashova-Sanz, Siegfried Wahl

**Affiliations:** 1Institute for Ophthalmic Research, University of Tübingen, 72076 Tübingen, Germany; siegfried.wahl@uni-tuebingen.de; 2Carl Zeiss Vision International GmbH, 73430 Aalen, Germany

**Keywords:** visual search, virtual reality, subtle visual augmentation, realistic visual scenes

## Abstract

Visual search becomes challenging when the time to find the target is limited. Here we focus on how performance in visual search can be improved via a subtle saliency-aware modulation of the scene. Specifically, we investigate whether blurring salient regions of the scene can improve participant’s ability to find the target faster when the target is located in non-salient areas. A set of real-world omnidirectional images were displayed in virtual reality with a search target overlaid on the visual scene at a pseudorandom location. Participants performed a visual search task in three conditions defined by blur strength, where the task was to find the target as fast as possible. The mean search time, and the proportion of trials where participants failed to find the target, were compared across different conditions. Furthermore, the number and duration of fixations were evaluated. A significant effect of blur on behavioral and fixation metrics was found using linear mixed models. This study shows that it is possible to improve the performance by a saliency-aware subtle scene modulation in a challenging realistic visual search scenario. The current work provides an insight into potential visual augmentation designs aiming to improve user’s performance in everyday visual search tasks.

## 1. Introduction

Visual search is one of the most common tasks in everyday life, be it when a person is looking for a friend in a crowd or when a doctor is analyzing an optical coherence tomography (OCT) scan from a patient [1]. Search becomes more challenging when the time to find the target is limited. For example, when a person is searching for the keys right before leaving the house, or when a surgeon is performing a meticulous manual task during the surgery using digital surgical microscope images [2]. In this study we focus on how performance in a visual search under limited time conditions can be improved.

An extensive amount of research has been done to investigate how people are searching for a target among distractors, and which neural mechanisms are laying behind, where numerous search task paradigms have been implemented (for reviews, see [3,4,5]). The difficulty of the visual search task depends on various factors, including how similar the target and the background are, how distinct the target is from the distractors, how complex the scene is, whether the observer has seen the scene already before, and many other aspects [3,4,5]. The human capacity to process visual content is limited, and mainly, in complex searches, it is crucial to select and prioritize visual information to complete the task. Attention is one mechanism that supports visual search and enables the searcher to find the target more efficiently [6,7,8]. The shift of attention is associated with eye movements and fixating attended locations [5,9]. Among numerous aspects impacting guiding of attention in visual search is stimulus-driven saliency of different elements of the visual scene. Specifically, the observer’s attention can be attracted by salient distractors, even though from a goal-oriented perspective, they are irrelevant. To which extent saliency plays a role when it comes to visual search strategy has been long debated. Some studies showed that top-down mechanisms primarily drive visual search strategy, where fixation density can be explained by saliency only for the first few fixations [10,11,12].

Other studies demonstrated that salient goal-irrelevant distractors could attract the observer’s attention, slowing down the search [13,14,15,16]. The large variability of the results reported in the literature supports the notion that a combination of factors affects human attention guiding when looking for a target [17]. Here we approach visual search assuming that salient regions can attract attention in visual search.

When it comes to saliency, there are many different definitions that one meets in the literature. A salient region, in its broad context, is an area of the visual scene. It has a high contrast with its surroundings in one or multiple feature dimensions, be it color, shape, spatial frequency, speed of motion, contextual meaning, location within the visual field of view. Even though saliency is often associated with bottom-up low-level feature contrast [18,19], both aspects are connected, but not identical [20]. Therefore, a part of a scene is considered salient if it is likely to attract the observer’s attention, which is usually accompanied by fixating on that region.

The vast majority of existing knowledge on visual search is based on experimental studies conducted on conventional 2D-screens often using not-realistic synthetic stimuli as search arrays in a controlled environment. The search behavior in such artificial conditions can differ from real-life scenarios. A group of studies investigated human’s visual behavior in a search task using naturalistic scenes displayed in a 2D-screen (e.g., [12,21,22,23,24,25]). The fast development of modern technologies such as Virtual Reality (VR) and VR eye tracking enabled researchers to study visual search in more realistic 3D environments [26,27]. In order to increase the level of immersion of the experimental paradigm, in the current study, we focus on real-world static visual scenes displayed in virtual reality, ensuring free body and head movement.

New technological tools, such as augmented reality (AR), enable the use of additional visual cues to purposefully guide user’s attention and improve user’s performance [28,29,30]. On the other hand, one drawback of augmenting visual input with additional content is that it introduces a trade-off of the potential benefit between performance and overlaying real visual scene with an additional layer of information. That, in turn, captures attentional resources, which essentially becomes a bottleneck for the design of visual augmentation [31]. In this study, we hypothesize that by subtly modifying the visual scene, it is possible to drive the observer’s attention away from salient distracting locations, enabling the user to find the target faster. Previously, some attempts were made to apply subtle visual content modification for gaze guidance, where color, luminance, spatial frequency, and other domains were modulated [32,33,34,35,36,37,38,39,40,41]. In this work, blur was selected as a domain for modifying visual content, as it was previously shown that blur, although with limitations, can be used for gaze guidance to an extent where the observer does not even notice the modification [42,43,44].

Furthermore, the idea of slightly defocusing parts of the scene for triggering the observer to fixate on more clear locations is widely used in photography and cinematography [45,46,47]. Also, Sitzmann et al. [48] proposed blurring images based on saliency to downsample the resolution of the non-salient regions for further image compression. In contrast to their approach, the strongest blur was applied to the most salient regions in this study.

This study investigates whether blurring salient regions of the visual scene, which would otherwise likely attract the observer’s attention, can improve visual search task’s performance, enabling the observer to find the target faster. Using a psychophysical approach, we evaluate observers’ ability to locate the target within a limited amount of time. Eye-tracking data was recorded to support the results. Specifically, fixations were analyzed.

## 2. Materials and Methods

### 2.1. Participants

Twenty naive participants (14 female and 6 male), with normal or corrected to normal vision were tested. Participants were aged between 20 and 38 years old. The study was conducted according to the guidelines of the Declaration of Helsinki. The study was approved by the ethics committee of the Faculty of Medicine at the University of Tübingen with a corresponding ethical approval identification code 138/2017b02. Signed informed consent was obtained from each subject before the measurements. All data were stored and analyzed in full compliance with the principles of the Data Protection Act GDPR 2016/679 of the European Union.

### 2.2. Experimental Setup

#### 2.2.1. Hardware Specifications

The visual content was displayed to the participant using HTC Vive Pro Eye (HTC Corporation, Taoyuan, Taiwan) virtual reality headset running on a Windows 10 PC with NVIDIA GeForce GTX 1070 graphics card (NVIDIA Corporation, Santa Clara, CA, USA). The field of view of the headset and the refresh rate reported by the manufacturer are 110° and 90 Hz, respectively. The participant interacted with the environment via the HTC Vive controller. The position and rotation of the headset and the controller were tracked via the HTC base stations 2.0. The eye-tracking data was collected using a built-in eye tracker at a frequency of 120 Hz.

#### 2.2.2. Software Specifications

The experimental paradigm was generated using the Unity Game engine [49], Unity version 2019.3.15.f1. The eye movement data was collected using Unity package SRanipal version 1.1.0.1. Recording of the eye movement data at a maximum sampling rate 120 Hz was realized by means of using a separate thread parallel to the main script execution [50]. The data analysis was performed using Python 3.6 packages NumPy [51] version 1.19.1, SciPy [52] version 1.5.2 and Pandas [53] version 1.1.3. The statistical analysis was conducted using R [54] version 3.6.1, in particular, package lme4 [55]. The data visualization was performed using Python packages Matplotlib version 3.3.1 [56] and Seaborn [57] version 0.11.0.

### 2.3. Virtual Environment and Stimuli

#### 2.3.1. Real-World Scenes

The virtual environment was composed of omnidirectional images displayed in virtual reality (VR) by back-projecting it to the Skybox sphere (Figure 1C). An omnidirectional image is a 360-degree panoramic image. An equirectangular projection is one way to represent an omnidirectional image, where the aspect ratio of the projection is 2:1. The horizontal and vertical coordinates of the projection are polar (ϕ) and azimuthal (θ) angles, respectively, where ϕ can be in the range from 0${}^{{}^{\circ}}$ to 360${}^{{}^{\circ}}$, and θ ranges from 0${}^{{}^{\circ}}$ to 180${}^{{}^{\circ}}$ (Figure 1A). For the main experiment, 24 scenes were selected from the Salient360! Training dataset with diverse content: indoor/outdoor, day/night time, containing people/not containing people, etc. (see full set of images in Appendix A). The resolution of the images ranged from 3000 × 1500 to 10,000 × 5000 pixels with mean resolution of (5878±2443) × (2939±1222) pixels. For the training phase, five scenes from the Salient360! Training dataset were selected, different from the scenes of the main experiment (see Appendix A).

#### 2.3.2. Saliency Maps

To evaluate which regions of the scenes are likely to attract attention, saliency prediction models are widely used [58]. In recent years interest in saliency models for 360-degree omnidirectional images largely grew [59]. In the current study, the saliency maps used for spatial modulation of the omnidirectional images were obtained using the method described in [60] which won in the “Head and Eye Movement Prediction” category of “Salient360!” Grand Challenge at ICME’2017 [61]. Authors of the method proposed composition of the continuity-aware and the cube map approaches using a combination of saliency predictors, with applied equator bias (for details, see [60]). The gray-scale saliency maps used for the main experiment can be found in Appendix A. The pixel color values of the saliency maps ranging from 0 (black) to 1 (white) are referred to as saliency values. In Figure 2 an example of one scene together with its saliency map is shown.

#### 2.3.3. Blurred Images

The omnidirectional images were modified by applying blur. The strength of blur was spatially varied based on the corresponding saliency map. In particular, each blurred image resulted from a convolution of the respective original image and a two-dimensional Gaussian kernel. The kernel’s size was fixed to one degree, whereas the standard deviation σ of the kernel was varied, determining the blur strength. A set of blurred 360-degree images using two different values of the Gaussian kernel standard deviation was generated. In particular, $\sigma_{1}=17\%$ of kernel size, and $\sigma_{2}=34\%$ of the kernel size, were used defining two of the experimental conditions (Figure 3). At a given experimental condition, the blur strength for each pixel was weighted with its saliency value. Specifically, the standard deviation of the Gaussian kernel for each pixel was computed by multiplying the standard deviation corresponding to the experimental condition ($\sigma_{1}$ or $\sigma_{2}$) with the respective saliency value of that pixel (values from 0 to 1, see Section 2.3.2). The size of the kernel in pixels, as well as the standard deviation, were scaled considering the individual image size in pixels. An example of a blurred part of an image is shown in Figure 3.

#### 2.3.4. Search Target

As a search target, a *Gabor cross* was used. Similar to [12], a Gabor cross is a sum of vertical and horizontal Gabor stimuli, where each of the Gabors is a product of 8 cyc/${}^{{}^{\circ}}$ cosine function with Gaussian envelope with standard deviations of 0.06${}^{{}^{\circ}}$ and 0.32${}^{{}^{\circ}}$ (Figure 4). The total width and height of the cross was approximately 3${}^{{}^{\circ}}$ visual angle. The target was overlaid with the visual scene by positioning it in Unity at a pseudorandom location on top of the omnidirectional image (see Section 2.3.5). To blend the target with the omnidirectional image, the transparency of the cross was decreasing towards the edges of the Gabor cross at a rate of the Gaussian envelope used to generate the cross.

#### 2.3.5. Search Target Locations

In each trial, the search target was positioned at a pseudorandom location defined by spherical coordinates (r,ϕ,θ). Specifically, *r* was set to a fixed value, whereas ϕ and θ were varied. Prior to the experiment, a set of nine possible target locations (ϕ,θ) was generated for each visual scene, where ϕ could be in the range [0${}^{{}^{\circ}}$, 135${}^{{}^{\circ}}$] and [225${}^{{}^{\circ}}$, 360${}^{{}^{\circ}}$], and θ in the range [45${}^{{}^{\circ}}$, 135${}^{{}^{\circ}}$]. This range was selected to ensure that subjects do not have to rotate their head in an uncomfortable position trying to look too much up or down. Additionally, taking into account that in two experimental conditions the visual scenes were spatially blurred based on the corresponding saliency values, the regions with low saliency were selected as possible search target locations to avoid displaying the target on very differently blurred backgrounds in different experimental conditions. Specifically, possible (ϕ,θ) were limited to regions with low saliency for each visual scene—locations with saliency values under 17% of maximum saliency value on corresponding saliency map. During the experiment, each participant had the same set of pseudorandom locations for corresponding visual scenes and respective conditions. The size of the search target was was kept constant in visual angle in all visual scenes. The distribution of the final set of possible search target locations for all visual scenes can be seen in Appendix A.

### 2.4. Experimental Procedure

#### 2.4.1. General Procedure

Each participant performed a visual search task wherein each trial they had to search for the search target located at a pseudorandom location within a limited amount of time set to 20 s. Participants were instructed to find the target as fast as possible, naturally moving their head and gaze. Once the target was found, participants were asked to fixate the target and press a button on the controller. As soon as the button was pressed, the trial was terminated. Participants performed the experiment in a seated position on a rotating stool enabling free head movement (Figure 1B). The experiment was performed in a single one-and-a-half-hour experimental session. First, each participant had a training phase where they got acquainted with the virtual environment and the search target. Next, the main part of the experiment was conducted. Every time the participant put on the headset, a five-point built-in calibration procedure of the eye tracker was performed. Each scene was explored starting from the same spatial point $\phi =180{}^{{}^{\circ}},\theta =90{}^{{}^{\circ}}$. To do so, before starting a new trial, participants had to fixate a reference fixation point of size 1${}^{{}^{\circ}}$ visual angle on a gray background located in the corresponding direction. The gaze position was controlled using the eye tracker. Once the gaze position condition was satisfied, the trial was started. That is, the scene with the search target was onset.

#### 2.4.2. Training Phase

The training phase consisted of 10 trials with no time limit. In each trial, one of five different visual scenes was shown with the search target overlaid on top of the scene. Thus, each scene was presented twice. The scenes used in the training phase were different from the ones used in the main experiment (see Appendix A). By the end of the training phase, all participants verbally responded that the task was straightforward and that they felt comfortable to continue with the main experiment.

#### 2.4.3. Main Experiment

In the main experiment, participants were instructed to perform the search task the same as during the training phase. Furthermore, they were informed that some scenes are partly blurred, but that it does not affect the task instructions. Participants were also informed that each trial’s time is limited to 20 s, but that in some trials, it is too difficult to find the target in the background. Thus, it is normal not to find the target within the given time in some trials. Each trial finished pressing a button if the target was found or after the time limit was reached. The fixation on the target was not controlled during the trial. The main experiment was split into three blocks with short three-to-five minutes breaks. During each break, the participant took off the headset and rested. Before starting each next block, the eye tracker calibration procedure was conducted. Each block consisted of 72 trials. All 24 scenes were presented in each block three times: as an original image with no blur, blurred with standard deviation $\sigma_{1}$, and blurred with standard deviation $\sigma_{2}$. The blur strength determined three experimental conditions: Condition 0 (no blur), Condition 1 (blurred with $\sigma_{1}$), and Condition 2 (blurred with $\sigma_{2}$). During each block, all three conditions were presented in random order. Each block’s duration varied depending on how long it took the participant to find the target in each trial, ranged between 10 and 20 min. During the main experiment, each participant performed a total of 216 trials divided into three blocks, with each block containing the three conditions above-mentioned for every scene (24). For each scene, the search target location was always different selected from the set of nine possible locations (see Section 2.3.5).

### 2.5. Analysis

The behavioral data as well as the eye movement data were analyzed using Python as well as lme4 library of R, and compared using linear mixed model analysis [62].

#### 2.5.1. Behavioral Performance Metrics

The *search time* and the *proportion of missed trials* were selected as the main behavioral performance metrics. The search time is defined as the time since a trial started until the button was pressed. Thus, it is a continuous variable ranging between 0 ms and 20,000 ms. Only trials where the target was found were used to compute the search time. The proportion of missed trials is computed as the number of trials where the target was not found within 20 s divided by the total amount of trials in the respective experimental condition. As for each condition, there were 72 trials from each participant. The proportion of missed trials is a discrete variable with a minimum step of 1/72=0.014. Better performance is defined by a shorter search time as well as a lower proportion of missed trials. The impact of blur was estimated by fitting the linear mixed models to the data, fitting a separate model for each of the metrics, where *search time* and *proportion of missed trials* are dependent variables, *blur condition* is fixed effects, and *subject* is a random factor. Furthermore, to evaluate learning throughout the experiment, the *block number* was introduced as an additional fixed effect in the model. To further explore whether the differences between blur conditions evolve with time, an extension of the model could be done where one could account for possibly different slopes in data subgroups at a given blur condition and given block number. However, the limited amount of data points collected in this study does not support a more complex model. Therefore, a proposed linear mixed model with two fixed effects and one random factor was selected. An extensive quantitative analysis of performance evolution over the course of time is out of the scope of the present work and is a subject of future studies.

#### 2.5.2. Eye Movement Metrics

As a secondary set of metrics to characterize visual search performance, the number of fixations until the target was found (*NumOfFix*), the proportion of fixations within the area of interest (*PropFixInAOI*), and duration of fixations (*FixDuration*) were computed. The area of interest (AOI) is defined, similar as in [48], as 5% most salient pixels of the original image based on its saliency map (see example in Figure 5). The proportion of fixations within the area of interest is defined as the number of fixations within the area of interest divided by the total number of fixations for each trial. The number of fixations until the target was found was defined as the total number of fixations in each trial where the target was found. Finally, the fixation duration was computed as the duration of fixations in each trial. The effect of blur condition was evaluated for each metric by fitting a separate linear mixed model where *NumOfFix*, *PropFixInAOI*, and *FixDuration* are dependent variables, *blur condition* is fixed effect, and *subject* is a random factor. To account for possible learning throughout the experiment, the *block number* was introduced as an additional fixed effect in the model.

#### 2.5.3. Eye Movement Raw Data Pre-Processing

The eye movement data were recorded at a frequency of 120 Hz. The gaze position data was accessed using a customized written Unity script utilizing the HTC SRanipal SDK package functions [63]. The time variable was taken using the system time, as it was previously shown that the time function from SRanipal package returns not always reliable values [64]. In Table 1 the main recorded variables are described. All variables were recorded for left and right eyes.

To prepare the data for further processing, first, similar to [64], the raw data were filtered based on the eye data validity bit mask value, which represents the bits containing all validity for the current frame. After the filtering, only the data where the eye data validity bit mask had value 31 for both eyes, were selected. Doing so, the data where the eye tracker partly or completely lost the pupil (including blinks) was filtered out. Next, the gaze position was calculated in spherical coordinates. In particular, the polar ϕ and azimuthal θ angles were computed using Equations (Equation 1) and (Equation 2). In Unity, the *z*-axis corresponds to the depth dimension.
(1)$$\phi =\arctan\frac{x}{z},$$
(2)$$\theta =\arctan2\;(y,\sqrt{x^{2}+z^{2}}),$$
where (x,y,z) are coordinates of normalized gaze directional vector in headset coordinates. Note that SRanipal returns the gaze direction vector in the right-handed coordinate system. To convert the coordinates in the left-hand coordinate system (same as Unity world coordinate system, see Figure 1B), the *x*-coordinate was multiplied by −1. To compute the gaze position in Unity world coordinate system, the gaze position in headset coordinate system was multiplied by the head rotation quaternion. In Figure 6 an example of gaze position for one subject in one trial in spherical coordinates (ϕ, θ) is shown.

#### 2.5.4. Fixation Detection Algorithm—I-VT

Fixations were identified using *velocity threshold algorithm for fixation identification* (I-VT) [65]. The algorithm was implemented following the description in [66,67]. The gaze velocity *v* was computed in ${}^{{}^{\circ}}$/s between each two consecutive samples (Equation Equation 3).
(3)$$v=\frac{\sqrt{{\left(\phi_{i}-\phi_{i-1}\right)}^{2}+{\left(\theta_{i}-\theta_{i-1}\right)}^{2}}}{t_{i}-t_{i-1}},$$
where $\left(\phi_{i},\theta_{i}\right)$ and $\left(\phi_{i-1},\theta_{i-1}\right)$ are consecutive gaze positions in degrees visual angle in headset coordinates, and $t_{i}$ and $t_{i-1}$ are respective time stamps. To reduce the noise level of the data, a running average filter was applied with the window size of three samples which is ∼25 ms. An eye movement was considered to be a fixation if the gaze velocity did not exceed a threshold 60 ${}^{{}^{\circ}}$/s [68]. Two fixations were merged in a single fixation if the time between them was under 75 ms [69], and the angular distance was under 1${}^{{}^{\circ}}$ [69,70]. Too short fixation with a duration under 60 ms were filtered out [69,70]. In Figure 7 the eye movement data processing algorithm is summarized in a flow chart.

## 3. Results

### 3.1. Behavioral Data

In Figure 8 the mean search time and the proportion of trials where the target was not found, estimated across all subjects, are shown. The mean value of the search time considering trials where the target was found was 8423 ms±1358 ms, 8146 ms±1258 ms, and 7670 ms±1550 ms for conditions 0, 1, and 2, respectively. The mean value of the proportion of trials where the target was not found, was 0.17, 0.11, and 0.10 for conditions 0, 1, and 2, respectively. The individual data for each subject can be found in Appendix A. From linear mixed model analysis (see Section 2.5.1) over the course of all trials, a significant effect of blur was found for both behavioral metrics. Specifically, for the search time, a significant difference between the no-blur and $\sigma_{2}$-blur conditions was found with p<0.001. The difference between the no-blur and $\sigma_{1}$-blur conditions was close to but not significant with p=0.09. In terms of the proportion of trials where the target was not found, significant differences between the no-blur and $\sigma_{1}$-blur, as well as between no-blur and $\sigma_{2}$-blur conditions, were found with p<0.001 for both conditions.

To check for the learning effect, the mean search time and proportion of missed trials were evaluated over the course of the experiment (see Appendix A). A significant effect of block number was found for mean search time as well as for proportion of missed trials with p<0.001. This can also be observed as a downtrend of both behavioral metrics for all three blur conditions indicating a general learning of the task. This is expected, as with time participants get more familiar with the VR headset, as well as get more acquainted with the visual scenes since the same images were presented in each experimental block. Despite general learning, the difference in performance metrics between the no-blur condition and other two blur conditions was observed already in the first block of the experiment, most prominently, for the proportion of missed trials.

### 3.2. Eye Movement Data

In Figure 9A distribution of fixation duration is visualized as a kernel density estimate plot which is an alternative to a histogram plot. The plot represents all fixations computed for all subjects and all trials regardless of whether the target was found or not in a respective trial. The total number of detected fixations was 21,504, 18,059, and 16,092 for the Conditions 0 (no blur), 1 ($\sigma_{1}$), and 2 ($\sigma_{2}$), respectively. The mean duration of fixation averaged over all subjects across all trials is 133 ms±76 ms. This value is comparable with typically reported fixation duration considering that in a visual search task, fixations are usually shorter than in a free-viewing task [71]. In Figure 9B an estimate of the number of fixations until the target was found (NumOfFix) is shown. To estimate NumOfFix only trials where the target was found were considered. The mean values of NumOfFix are 11.9, 10.8, and 9.6 for Conditions 0, 1, and 2, respectively. Figure 9C demonstrates an estimate of the proportion of fixations within the area of interest (PropFixInAOI). To estimate PropFixInAOI, all trials were considered regardless of whether the target was found or not in the respective trial. The mean values of PropFixInAOI are 0.23, 0.20, and 0.19 for Conditions 0, 1, and 2, respectively. Finally, in Figure 9D an estimate of fixation duration (FixDuration) is shown. To compute the estimate of FixDuration all fixations were considered regardless of whether the target was found or not and whether a fixation was within AOI or not. See details of metrics’ definitions in Section 2.5.2. The individual data for each subject can be found in Appendix A.

From the linear mixed model analysis (see Section 2.5.2) over the course of individual trials and fixations, a significant effect of blur was found for NumOfFix and PropFixInAOI for all blur conditions. Specifically, for both metrics (NumOfFix and PropFixInAOI), a significant difference between the no-blur and $\sigma_{1}$-blur, as well as between the no-blur and $\sigma_{2}$-blur conditions, were found with p<0.001 for both conditions. For FixDuration a significant difference was found between the no-blur and $\sigma_{2}$-blur conditions with p<0.05. The FixDuration difference between the no-blur and $\sigma_{1}$-blur conditions was not significant with p=0.3. The differences in FixDuration only for fixation within AOI were not significant with p=0.61 and p=0.16 for Condition 0 vs Conditions 1, and Condition 0 vs Condition 2, respectively.

Evolution of PropFixInAOI was assessed across the time course of the experiment to check for learning (see Appendix A). A significant effect of block number was found with p<0.001. This indicates a general learning effect across three blocks of measurements which as mentioned in the previous section, can be expected due to familiarizing of subjects with the setup and the scenes. Nonetheless, the proportion of fixations in the AOI in the blurred conditions compared to the non-blur condition appears to be smaller already in the first block of the experiment.

## 4. Discussion

In a visual search task, it was investigated whether blurring salient regions of the visual scene can drive the observer’s attention away from those regions and enable the observer to find the target faster. Using a psychophysics approach implemented in VR, we evaluated the observer’s ability to find the target in a real-world scene within a limited amount of time, as well as how long it took the observer to find the target. The eye movement data were evaluated by accounting for the number of fixations within the most blurred areas and the total number of fixations until the target was found. Similarly, the mean duration of fixations were computed for a non-blur and two blur conditions.

Overall, the experimental paradigm captured well a challenging visual search task in a realistic 3D environment. The possibility of freely moving head and gaze brought the controlled experimental setting closer to a real-life scenario compared to traditional screen-based paradigms.

The significant decrease found in the search time in conditions when salient regions were blurred shows that modulating the visual scene by spatially applying saliency-aware blur can lead to a faster locating of the target. In particular, in the condition with the strongest blur ($\sigma_{2}$), the gain in search time was almost 10% compared to the no-blur condition. A more noticeable difference was found for the proportion of trials where the target was not found. Specifically, compared to the no-blur condition, in both blur conditions ($\sigma_{1}$ and $\sigma_{2}$), participants missed fewer trials with a significant drop of 40%. These results indicate a possibility to improve performance in a challenging visual search task via partial scene modulation using blur.

Further analysis of the eye movement data, in particular, fixations, revealed a significant decrease of both: the number of fixations until the target was found, and the proportion of fixations within the area of interest. These results correlate well with the trend observed in the behavioral metrics. Specifically, participants needed to make fewer fixations to find the target in blurred conditions than the non-blur condition, resulting in shorter search times and fewer missed trials. The fact that participants fixated fewer times within the salient areas when those areas were blurred supports our hypothesis that the observer’s gaze was driven away from the scenes’ blurred locations, contributing to a more successful search in terms of search time and ability to find the target. One possibility is that participants learned over the course of the experiment that the target is located in not blurred areas, which could result in a tendency to search non-blurred regions. However, the emerging difference in performance metrics and PropFixInAOI between the no-blur and two blur conditions already in the first block of the experiment illustrates that, although some learning took place indicated by a significant effect of block number, participants did not simply learn to search only non-blurred regions. These findings are in line with existing knowledge on eye movements during real-world scene viewing. Several authors [23,72,73] showed that when a low-pass spatial frequency filter is applied across the visual field, saccades are preferentially initiated to unfiltered scene regions, both in free-viewing memorization task and visual search task. Precisely, the authors showed that when a low-pass filter, or blur, is applied to the central field of view, saccadic amplitudes tend to increase, and the number of fixations decreases. Accordingly, our results also indicate that fewer saccades landed in blurred areas, and subsequently, fewer fixations fell in those regions. In accordance with the literature available, our results support the notion that eye movements are adjusted to increase the potential usefulness of inspected visual information, moreover when foveating higher spatial frequency regions of the scene [72,73,74,75,76,77,78,79].

Fixation duration, however, did not change much across different blur conditions. Cajar et al. [72] suggested that, concerning the central field of view, fixations were longer when the available spatial frequencies matched the necessary function of foveal vision, which is the analysis of details, compared to the low-pass filtered central field. On the other hand, Nuthmann [77] argued that fixations become longer when the task requires higher processing capacity, for example, when the image is slightly blurred, and it is more difficult to distinguish details. Contrarily, Cajar et al. [72], found that fixations increase in duration when the visual task becomes more difficult due to spatial-frequency filtering only when the task complexity is moderate, that is when the viewer still can make use of fixating on some locations of the scene for a longer time. In the current study, different parts of the scene were blurred to a different extent. We suggest that observers fixated mostly on more sharp areas, as is also evident from the data (PropFixInAOI), and did not need to adjust their fixation duration significantly. It is important to note that the task nature and the type of search target also play a role in fixation duration [12,71,80]. In the current study, the search target was a simple Gabor-cross in contrast to some sophisticated objects. The target was always known and was constant throughout the whole experiment. Therefore, the fact that fixation duration did not change much across the different experimental conditions indicates that the task difficulty did not vary significantly, but participants just needed fewer fixations to locate the target.

When studying visual search in VR, several challenges arise, producing some limitations in the current study. One aspect is the variability in the target visibility across the different combinations of scene-target location. It is easier to find the target in some trials because it is more visible in the background, whereas it might be more difficult in others. The target’s visibility depends on many factors such as contrast, spatial frequency, and retinal eccentricity, as also discussed by Rothkegel et al. [12]. Simultaneously, the difficulty of finding the target represented by how long it took the participant to locate it in each trial depends not only on the visibility of the target but also in which direction (clockwise or counterclockwise) the observer started the search relative to the target location. In the current study, we did not control for the target visibility. Furthermore, by keeping the search target abstract rather than a meaningful object, we avoided spatial scene-contextual bias of the search where observers would likely fixate only the scene’s locations containing anchors for the searched object [26]. However, in real life, we search for real objects. Thus, it is of interest to test our approach on realistic objects as the search target, although that may induce further bias, as some objects are expected in specific locations but not others. Also, in our study a high spatial frequency target was used. By blurring some parts of the image the distribution of the spatial frequencies of visual scene shifts towards lower frequencies. Considering feature-based guidance, this could cause the searchers to fixate less on low pass filtered locations resulting in a better performance in blurred conditions. Testing more search targets also of low spatial frequencies could provide further insight into the relation between the spatial frequency distribution of the scene and the target. In the present study, however, by selecting a certain range of possible target locations, the immediate proximity of the target was not altered by blurring. Based on the results of the study, blurring salient locations potentially can improve visual search performance at least when the search target is of high spatial frequency. Another particularity of the current study is the variability of visual scenes. While the different scenes eliminate the spatial bias of the scenes’ search and learning, it introduces a scene dependence of the salient areas. Nonetheless, it is reasonable to use multiple diverse, realistic scenes for the visual search task [12,21], as that is what we observe in real life. One more aspect to keep in mind is that we based the spatial blur pattern on a saliency map computed by a model in this study. The ultimate goal of using saliency maps, as aforementioned, is to know which locations are likely to be fixated by the viewer. The vast majority of fixation prediction models, including those we used in this study, are developed for a free-viewing task and not a search task. Therefore, for further improvement of our method, it would be interesting to generate saliency maps using more fitting models, among which can be deep-neural-network based models trained not only on ground-truth spatial locations of the fixations but also taking into account the temporal dimension of fixations [59,81]. In addition to fixation analysis, it is of interest to evaluate saccadic behavior. But compared to a screen-based paradigm, it becomes more challenging to detect saccades in the eye-tracking data stream due to a high noise level caused by the headset and rapid head movement. Thus, further development of methods for a more reliable saccade detection in VR eye-tracking data is necessary such as, for example, proposed by Diaz et al. [82]. Also, when applying a blur to the visual scene, one has to keep in mind that the visual information is partly impaired in blurred regions. This study shows that even a small blur ($\sigma_{1}$) can improve visual search performance. However, when the task is more demanding in processing small details, the blur strength may affect the task. Future studies analyzing how much blur can be applied improving performance in search tasks while not disturbing other performance are required. Finally, it is worth mentioning that the levels of blur used in this study were selected based on a subjective judgment of the experimenters. Specifically, a relatively small ($\sigma_{1}$) and a medium ($\sigma_{2}$) blur levels were chosen. Although no systematic analysis of subjective blur perception was conducted, some subjects reported that they noticed some parts of the scenes to be blurred more than others. Beyond the scope of the present work, future studies on subjective blur awareness can be conducted (e.g., [42]).

In the present study, a setting was used, where the target was located in not salient, and consequently, not blurred regions. This way it was intended to avoid a potential pop-out effect of the target when blurring the background. In the scope of this study, the results support the notion that blurring most salient regions can make the search more efficient at least when the target is situated in not salient locations. For subsequent studies it would be interesting to investigate a broader range of settings such as blurring non-salient regions, or blurring locations not based on saliency.

Regarding the guiding mechanisms in visual search, there are multiple factors which affect the search to a different extent, including bottom-up saliency, top-down goal-oriented feature guidance, scene guidance, recent search history of the observer as well as effects of value [4]. Various models of attentional guidance in visual search have been proposed describing the interplay of those factors (for review see [83]). A commonly used framework to describe the neurophysiological mechanism of combining the guidance elements is a priority map which is a weighted average of different factors. The priority map is considered to combine the representation of bottom-up object’s distinctiveness and its top-down relevance to the observer [84,85]. The impact of saliency in visual search guidance has been debated, where the contribution of saliency is apparent in some tasks at hand [13,14,15,16], but not in others [10,11,12]. The neurophysiological studies also demonstrate a varied saliency contribution in the priority map building across different tasks [84]. In the present study, by demonstrating more efficient search upon suppression of salient regions, our findings, among others, implicitly show that saliency can indeed play a role in visual search strategy not only in case of synthetically generated search sets, but also in real-world visual scenes. Importantly, results of the current work illustrate saliency impact in a 360-degree scenario where a natural free head rotation is enabled as well the visual field of view is significantly larger in contrast to screen-based studies. Even so, from previous studies it is clear that the role of saliency in visual search guidance is limited [4]. Such, if the search target would be more naturalistic and context-dependent, a smaller impact of saliency and a larger contribution of goal-oriented factors such as scene context, search history or value-based guidance can be expected. Nevertheless, we believe that through a 360-degree setting, multiple diverse visual scenes and use of a recent promising saliency model, our study contributes to understanding of saliency role in a real-world scenario of visual search.

This study served as a proof-of-principle of using a scene modulation by blurring salient regions to make the search more efficient. On a long term, the concept could potentially be implemented using, for example, see-through augmented reality head-mounted displays where the real-time video content could be modified. Another potential realization of the approach could be implementation into VR experiences such as VR simulations for professional training, e.g., in surgical procedures [86], driving simulators [87], or flight simulators [88].

## 5. Conclusions

To conclude, this work shows that, in a challenging realistic visual search scenario, it is possible to improve the task’s performance by a saliency-aware scene modulation, specifically, partial blur. The approach’s subtle nature is prone to support the user’s search strategy and not be disturbing. This study provides insight into potential visual augmentation designs aiming to improve user’s performance in challenging everyday visual search tasks.

## Figures and Tables

**Figure 1 brainsci-11-00283-f001:**
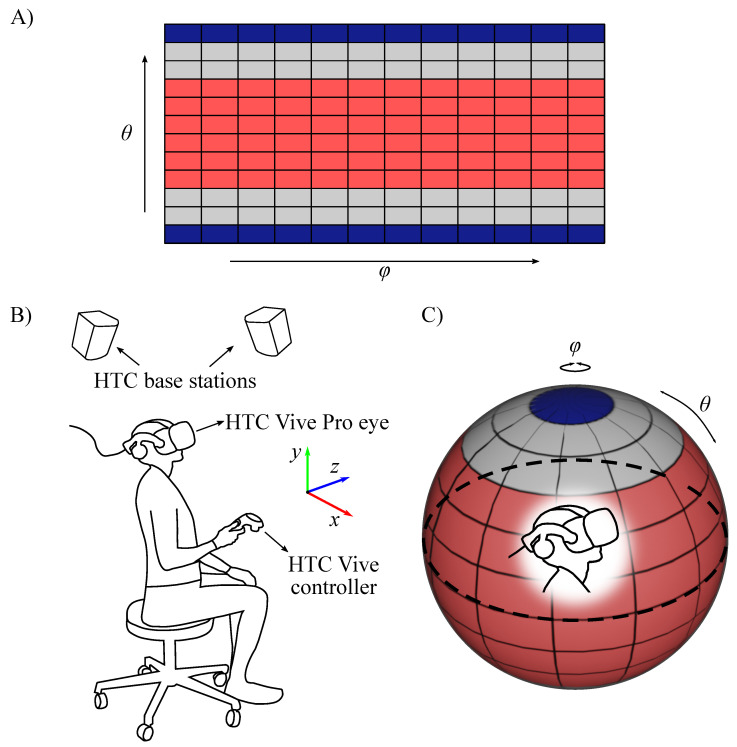
(**A**) A schematic representation of equirectangular projection. Each rectangle is 30${}^{{}^{\circ}}$ wide and 15${}^{{}^{\circ}}$ high. (**B**) The experimental setup. The experimental paradigm was implemented in Unity software and displayed in a VR headset—HTC Vive Pro Eye with a built-in eye tracker. Position and rotation tracking of the headset was realized via the HTC base stations 2.0. The participant performed the experiment in a seated position on a rotating stool. As soon as the search target was located, the participant pressed a button on the HTC Vive controller. Additionally, a left-hand coordinate system is shown, which is used to set the Unity world coordinates. For details on the experimental procedure see Section 2.4.1. (**C**) A schematic representation of a sphere to which the equirectangular projections (omnidirectional images) were back-projected in Unity as a Skybox. The colors indicate like-colored locations on the equirectangular projection in (**A**).

**Figure 2 brainsci-11-00283-f002:**
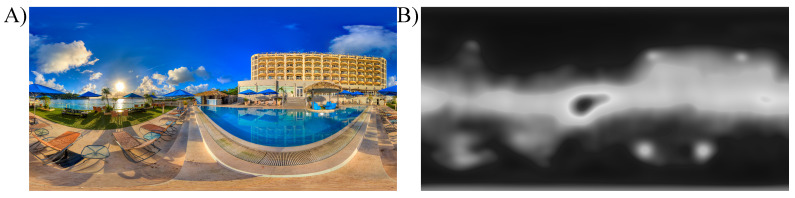
(**A**) Example of one omnidirectional image used in the main experiment. (**B**) The corresponding saliency map generated using the approach in [60]. White regions correspond to the most salient areas, whereas black color indicates the least salient locations.

**Figure 3 brainsci-11-00283-f003:**
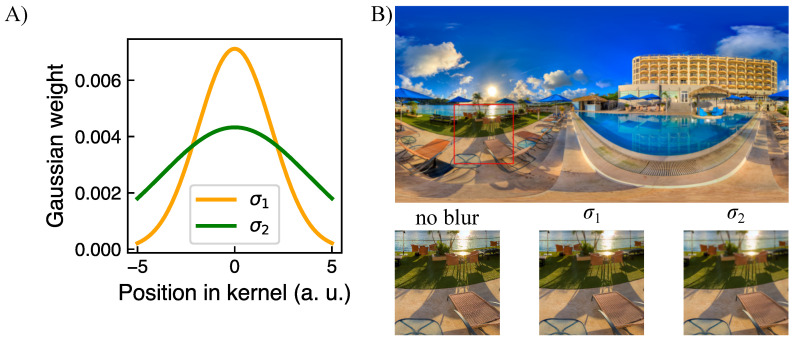
(**A**) A cross-section of a 2D Gaussian kernel used to blur images, with two standard deviations: $\sigma_{1}=17\%$ of kernel size, and $\sigma_{2}=34\%$ of the kernel size. The kernel size was fixed to a number of pixels corresponding to 1${}^{{}^{\circ}}$ visual angle. For schematic illustration, the kernel size in the plot equals 10 a. u. (**B**) Example of blurred part of one scene generated as a convolution of respective 2D Gaussian kernel and original image, considering its saliency map. In the lower row of (**B**), a part of the original image which is indicated by a red rectangular is shown in three blur conditions: no blur, Gaussian blur with maximum standard deviation $\sigma_{1}$, and Gaussian blur with maximum standard deviation $\sigma_{2}$.

**Figure 4 brainsci-11-00283-f004:**
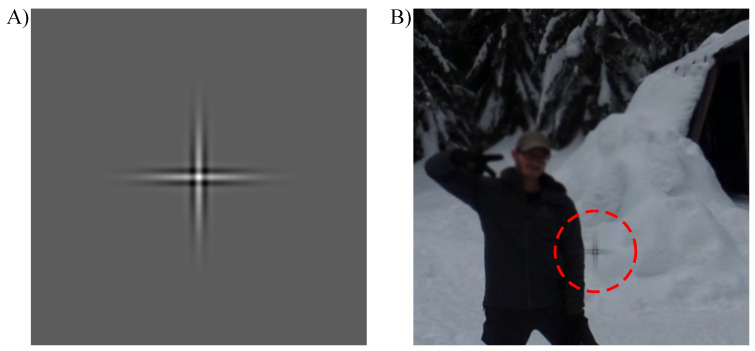
(**A**) The Gabor cross was used as a search target in the visual search task. It is a sum of vertical and horizontal Gabor stimuli, where each of the Gabors is a product of 8 cyc/${}^{{}^{\circ}}$ cosine function with Gaussian envelope with standard deviations of 0.06${}^{{}^{\circ}}$ and 0.32${}^{{}^{\circ}}$. (**B**) An example of the search target is overlaid with a visual scene at a pseudorandom location. For simplicity, the cross is highlighted by a surrounding red dashed line. In the experiment, the red line was not present.

**Figure 5 brainsci-11-00283-f005:**
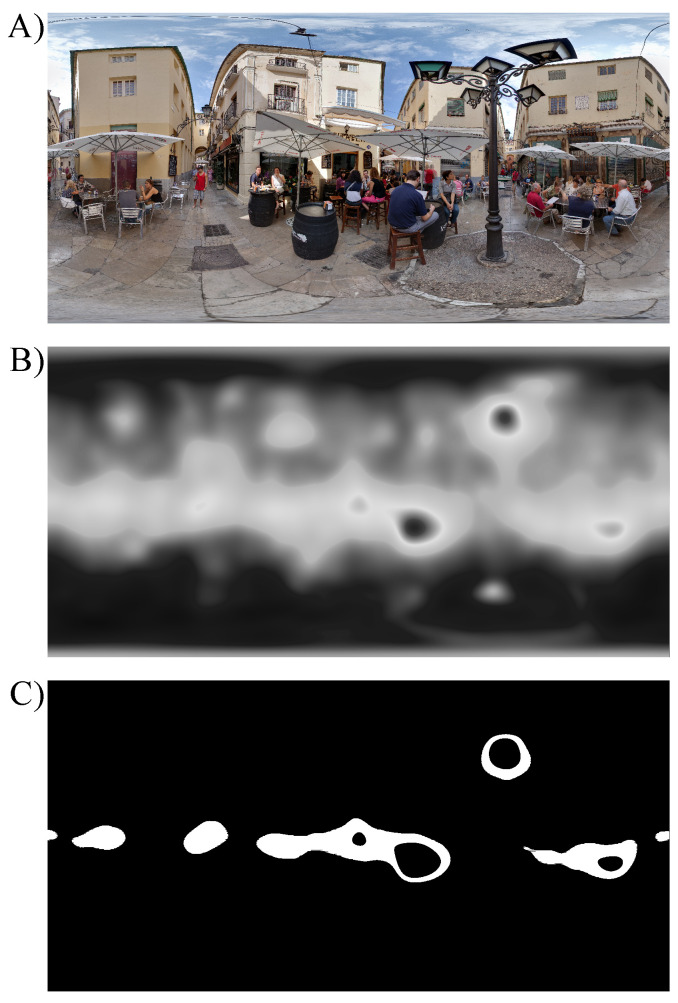
(**A**) Example of one omnidirectional image, (**B**) its saliency map in gray scale, and (**C**) selected 5% most salient pixels indicated by white color. The area of interest is the white region in (**C**).

**Figure 6 brainsci-11-00283-f006:**
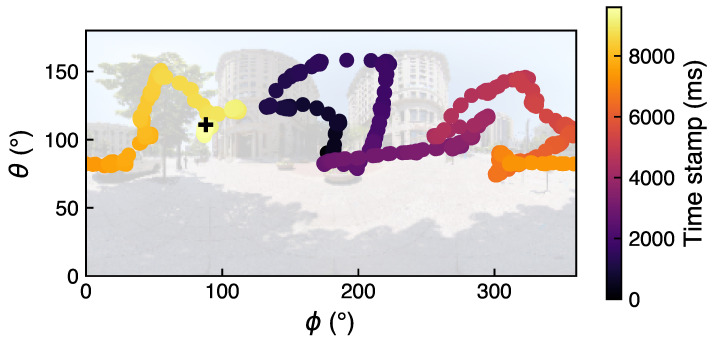
Example of raw gaze position in spherical coordinates for one subject in one trial. The background image is the visual scene shown in this particular trial—for easier visualization of the gaze positions, the omnidirectional image is washed out in the figure. The black cross with coordinates $\left(88{}^{{}^{\circ}},111{}^{{}^{\circ}}\right)$ indicates position of the search target in this trial. The color of the scatter points range from dark violet to light yellow and indicates the time stamp of each gaze position sample starting from the beginning of the trial: here, the participant started the search from the center of the image and ended around the target position. This trial lasted approximately ten seconds and the search target was successfully found by the participant.

**Figure 7 brainsci-11-00283-f007:**
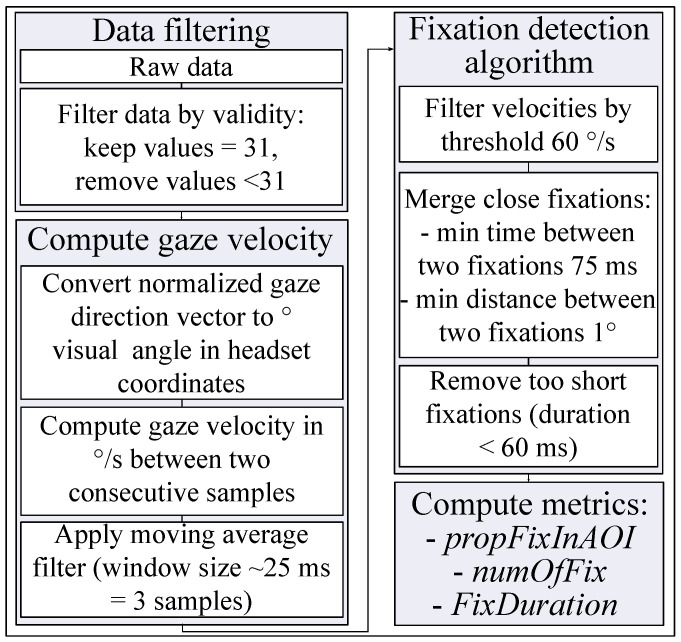
Eye movement data processing algorithm. For details see Section 2.5.4.

**Figure 8 brainsci-11-00283-f008:**
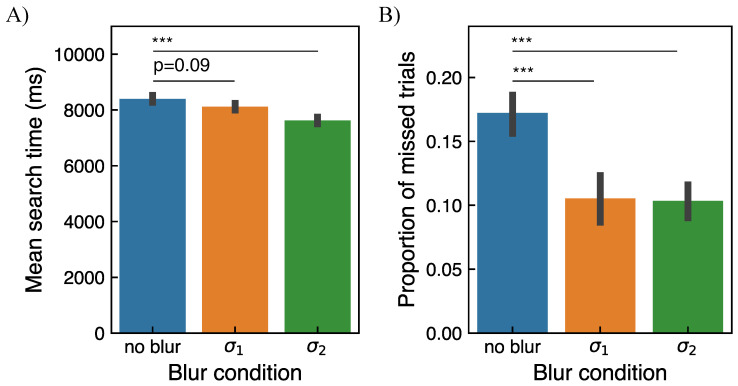
(**A**) An estimate of mean search time in each blur condition. The mean search time was computed for each subject by averaging the search time over the number of trials where the target was found. (**B**) An estimate of the proportion of trials where the target was not found in each blur condition. The bar plots represent data from 20 subjects. The error bars show the standard error of the mean. The indicators of significant differences obtained from the linear mixed model analysis are *: p<0.05, **: p<0.005, ***: p<0.001. For non-significant differences, the *p*-value is shown.

**Figure 9 brainsci-11-00283-f009:**
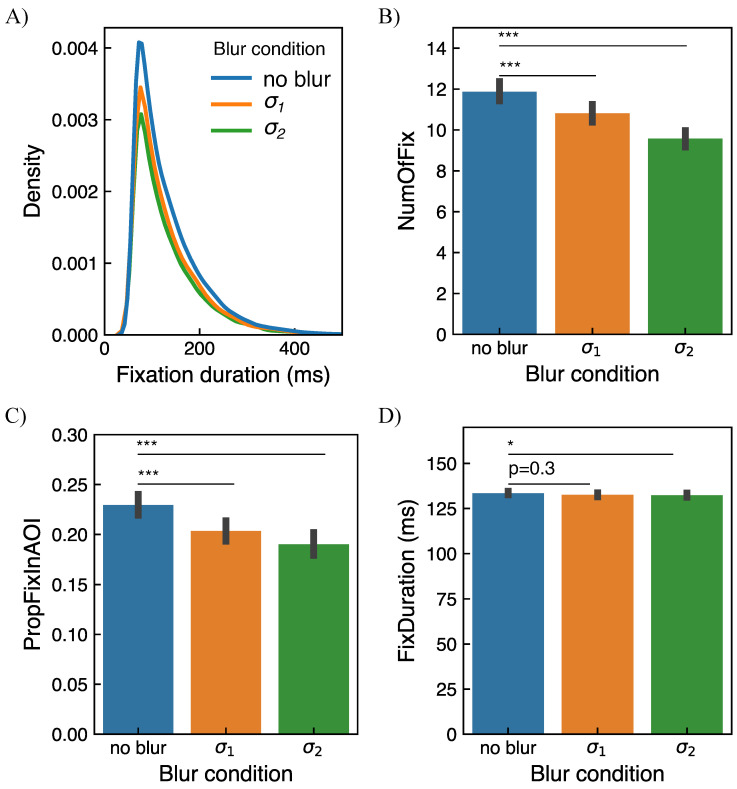
(**A**) A distribution of fixation duration as a kernel density estimate plot. Only fixations with duration up to 500 ms are shown to better resolve differences between different blur conditions. The mean duration of fixation averaged over all subjects across all trials is 133 ms±76 ms. The plot represents all fixations computed for all subjects and all trials. Each curve is normalized to the number of observations such that the total area under all densities sums to 1. (**B**) An estimate of the number of fixations until the target was found in each blur condition. The bar plots represent data from 20 subjects considering only trials where the target was found. (**C**) An estimate of the proportion of fixations within the area of interest in each blur condition. The bar plots represent data from 20 subjects considering all trials. (**D**) An estimate of fixation duration in each blur condition. The bar plots represent data from 20 subjects considering all trials. The error bars show the standard error of the mean. The indicators of significant differences obtained from the linear mixed model analysis are *: p<0.05, **: p<0.005, ***: p<0.001. For non-significant differences, the *p*-value is shown.

**Table 1 brainsci-11-00283-t001:** Main eye- and head-movement-related raw variables recorded during the experiment.

Variable	Units	Meaning
Timestamp	any integer number	The system time in ms at the moment of sample recording.
Eye data validity bit mask	an integer from 0 to 31	Indicates the validity of the data. A value of 31 indicates the highest validity of the recorded data. This parameter is used to filter the raw data where the eye tracker lost the pupil, including filtering blinks.
Gaze normalized direction vector	A three-coordinates vector *(x, y, z)* with each coordinate ranging from −1 to 1	A gaze vector indicating the direction of gaze in the headset right-hand coordinate system. To convert it to the left-hand coordinate system (Figure 1B), the x coordinate was multiplied by −1.
Head rotation	a rotation quaternion *(x, y, z, w)* of head	A quaternion describing the rotation of the headset in Unity world coordinates. The position of the headset was always fixed to the origin (0, 0, 0).

## Data Availability

Publicly available datasets were analyzed in this study. This data can be found here (accessed on 14 December 2020): osf.io/83kdh.

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
