# Peer review of "Saliency-Aware Subtle Augmentation Improves Human Visual Search Performance in VR"

_brainsci, 2021, doi:10.3390/brainsci11030283_

Round 1
Reviewer 1 Report
In this paper, the authors present an approach to improve visual search task's performance by blurring salient regions of the visual scene. The introduction and related work are well written, providing good motivations and sufficient context for the studies. The approach is described very well, an experiment with 20 participants were conducted and the findings are shown.
The vast majority of existing knowledge on visual search is based on experimental studies on conventional 2D-screens using synthetic stimuli. The novelty of the approach is the use of modern VR technology equipped with eye-tracking and panoramic images presenting 24 different scenes.
The studies are described in detail and easy to understand. The data collected and the analyses made is well described. The results are presented in bar plots and discussed extensively and provides insights into visual augmentation to improve performance in everyday visual search tasks.
The authors
Supplemental material shows the panoramic images used and the individual results of each participant.
The writing and figures are clear in all sections.
I found the proposed concept of using VR for the study a novel and promising idea. For future investigations the application of panoramic images can be enhanced by realistic 3D modeled environments where selective sharpness of depth can be applied to improve everyday visual search tasks.
The authors do not provide any information how to transfer the findings to a real-life application (special headsets, glasses, ...) and usage context.
Some remarks:
page 5, line 148: ... visusal angle. -> visual
page 6, caption of Figure 3, last line: with maximum standard deviation ....;
-> "with standard deviation s1, and Gaussian blur with standard deviation s2."
I understood that the same standard deviation sigma1 and sigma2 is used to blur the salient regions, not a maximum standard deviation s1 and maximum s2. what would be then the minimum, average, ...
Figure 7: more distance between the two columns makes the diagram clearer and shows the continuous sequence better.
Reviewer 2 Report
Lukashova-Sanz and Wahl investigated how blurring of salient image regions affects visual search in a VR setting. To this end, they placed Gabor crosses as search targets in omnidirectional images and applied different levels of blur to salient image regions. The results indicate better search performance (shorter search time, lower number of fixations, fewer misses) with higher amounts of blur.
The research question is interesting and the setup is certainly innovative and highly advanced. While I think that the study has some value as a methodological proof-of-principle, I don’t think that the experimental design allows to draw specific conclusions that would further our understanding of visual search. Therefore, I think that the study would find a more suitable audience at a more technically-oriented journal.
In order to conclude that the observed results are specific for the blurring of salient regions, one would need a control experiment in which non-salient regions are blurred. Furthermore, an important alternative interpretation of the results might be that visual search for a high spatial-frequency target is facilitated by reducing power at high spatial-frequencies (i.e. blurring parts of the image). To understand whether blurring is effective because it reduces the saliency of the affected image regions or because it increases the difference between the target and the background, one would need to cross the type of the target (low or high spatial-frequency) with the type of the power reduction in the image (low or high spatial-frequency).
Another potential issue to discuss is whether participants might have learned that the target never appeared in the salient/blurred regions and therefore tended to search in low-salient/unblurred regions. This could be checked by analysing how search performance and proportion of fixations in the AOI changed over the course of the experiment.
Minor comments:
I’m curious, were the participants aware of the blur in the image? Looking at Figure 3B, it seems that at least the level 2 blur might have been quite visible.
Since individual search trials were quite long, it might be interesting to analyse if the proportion of fixations in the AOI decreased over time (as has been shown in some cases in 2D scenes).
Line 54: I think this statement is not quite correct, there are numerous studies that investigated visual search in natural or man-made scenes (e.g. Drewes et al., 2011; Eckstein et al., 2017).
Line 67: Again, I think there are more examples for gaze guidance in VR (e.g. Danieau et al., 2017; Pomarjanschi et al., 2012; Sridharan et al., 2015).
References
Danieau, F., Guillo, A., & Doré, R. (2017, March). Attention guidance for immersive video content in head-mounted displays. In 2017 IEEE Virtual Reality (VR) (pp. 205-206). IEEE.
Drewes, J., Trommershäuser, J., & Gegenfurtner, K. R. (2011). Parallel visual search and rapid animal detection innatural scenes. Journal of Vision, 11(2):20, 1–21.
Eckstein, M. P., Koehler, K., Welbourne, L. E., & Akbas, E. (2017). Humans, but not deep neural networks, often miss giant targets in scenes. Current Biology, 27(18), 2827-2832.
Pomarjanschi, L., Dorr, M., & Barth, E. (2012). Gaze guidance reduces the number of collisions with pedestrians in a driving simulator. ACM Transactions on Interactive Intelligent Systems (TiiS), 1(2), 1-14.
Sridharan, S., Pieszala, J., & Bailey, R. (2015, September). Depth-based subtle gaze guidance in virtual reality environments. In Proceedings of the ACM SIGGRAPH Symposium on Applied Perception (pp. 132-132).
Round 2
Reviewer 2 Report
I agree with most of the author responses and think that the revised manuscript is improved considerably. However, I still have a few further recommendations:
Learning: I agree that there is already a difference between the blur conditions in the first block, but this difference seems to increase in the later blocks (especially Figure S12 A & B). I think a full statistical analysis, including the interaction between blur condition and block number would allow for a more complete interpretation of what is going on here.
Awareness of blur: Although this has not been assessed systematically, I recommend that the authors disclose somewhere in the manuscript that some of the participants noticed the different amounts of blur in the images.
Line 435: I would remove “cognitive” because the following factors are not exclusively cognitive.
Line 444: I think “but not the others” should read “but not in others”.
Line 451: saliency role -> role of saliency.
